# Risk of dispersion or aerosol generation and infection transmission with nasopharyngeal and oropharyngeal swabs for detection of COVID-19: a systematic review

Arnav Agarwal [1,2] Shannon M Fernando [3,4] Kimia Honarmand [5]
Layla Bakaa,[6] Sonia Brar,[7] David Granton,[8] Dipayan Chaudhuri,[9] Devin Chetan,[10,11]
Malini Hu,[8] John Basmaji,[5] Fiona Muttalib,[12] Bram Rochwerg,[2,8]
Neill K J Adhikari,[13,14] Francois Lamontagne,[15,16] Srinivas Murthy,[17]
David S Hui [18,19] Charles D Gomersall,[20] Samira Mubareka,[21,22] Janet Diaz,[23,24]
Karen EA Burns,[14,25,26] Rachel Couban,[2,27] Per O Vandvik[28]

For numbered affiliations see end of article.

**Correspondence to**
Dr. Per O Vandvik;
per.vandvik@gmail.com

## ABSTRACT

**Objectives** SARS-CoV-2-related disease, referred to as COVID-19, has emerged as a global pandemic since December 2019. While there is growing recognition regarding possible airborne transmission, particularly in the setting of aerosol-generating procedures and treatments, whether nasopharyngeal and oropharyngeal swabs for SARS-CoV-2 generate aerosols remains unclear.

**Design** Systematic review.

**Data sources** We searched Ovid MEDLINE and EMBASE up to 3 November 2020. We also searched the China National Knowledge Infrastructure, Chinese Medical Journal Network, medRxiv and ClinicalTrials.gov up to 29 March 2020.

**Eligibility criteria** All comparative and non-comparative studies that evaluated dispersion or aerosolisation of viable airborne organisms, or transmission of infection associated with nasopharyngeal or oropharyngeal swab testing.

**Results** Of 7702 citations, only one study was deemed eligible. Using a dedicated sampling room with negative pressure isolation room, personal protective equipment including N95 or higher masks, strict sterilisation protocols, structured training with standardised collection methods and a structured collection and delivery system, a tertiary care hospital proved a 0% healthcare worker infection rate among eight nurses conducting over 11 000 nasopharyngeal swabs. No studies examining transmissibility with other safety protocols, nor any studies quantifying the risk of aerosol generation with nasopharyngeal or oropharyngeal swabs for detection of SARS-CoV-2, were identified.

**Conclusions** There is limited to no published data regarding aerosol generation and risk of transmission with nasopharyngeal and oropharyngeal swabs for the detection of SARS-CoV-2. Field experiments to quantify this risk are warranted. Vigilance in adhering to current standards for infection control is suggested.

## Strengths and limitations of this study

► A comprehensive literature search incorporating English and Chinese-language peer-reviewed literature, grey literature and preprint databases was conducted.

► The literature search included inpatient and outpatient testing settings.

► The literature search accounted for available direct and indirect evidence.

► Findings of this systematic review were limited to one study evaluating transmission risk to healthcare personnel conducting nasopharyngeal swabs; no studies directly quantified the risk of airborne transmission.

► The findings of the systematic review preclude substantive evidence-informed guidance, and warrant vigilance in adhering to current standards for infection prevention and control measures.

## BACKGROUND

In late December 2019, a novel coronavirus, subsequently referred to by the World Health Organization (WHO) as SARS-CoV-2, was identified as the cause of atypical pneumonia cases detected in Wuhan, China.[1] Since then, SARS-CoV-2-related disease (named COVID-19) has emerged as a global pandemic.[2] In 1 year, SARS-CoV-2 has spread to over 200 countries, infecting over 100 million individuals, and causing over 2 million deaths internationally.[3]

International infection control guidance initially stated that viral transmission is likely primarily through direct, indirect or close contact with infected salivary and respiratory

BMJ

droplets (5–10 µm in diameter).[4] More recently, the WHO and the Centers for Disease Control and Prevention have recognised that there is growing evidence regarding SARS-CoV-2 transmission from airborne exposure, particularly in poorly ventilated enclosed settings with extended exposures, and in the setting of aerosol-generating procedures in healthcare settings.[5–8]

Testing for COVID-19 is commonly conducted using nasopharyngeal and oropharyngeal swabs of the posterior wall of the pharynx. The risk of aerosol generation associated with obtaining nasopharyngeal or oropharyngeal swabs in patients infected with COVID-19 remains unclear. The potential to induce coughing may compound this associated risk, the extent to which is unknown.

Given thousands of nasopharyngeal and oropharyngeal swabs are conducted daily internationally, the risk of potential airborne transmission to otherwise uninfected patients and healthcare workers must be balanced with severe resource constraints in personal protective equipment. This dictates a need for an updated evidence synthesis and guidance regarding the risk of aerosolisation associated with swab testing. We therefore conducted a systematic review of the literature to determine the risk of aerosol generation and associated transmission associated with nasopharyngeal and oropharyngeal swabs for detection of SARS-CoV-2.

## METHODS

Prior to initiating this systematic review, WHO personnel reviewed and approved internal protocols for this systematic review. Given time constraints of the initial commissioned review for the WHO in May 2020 (7 days to completion), the protocol was not publicly registered or published.

### Literature search

We conducted a comprehensive search of Ovid MEDLINE and Embase with the assistance of a health information specialist (RC) from inception to 3 November 2020 (updated), using a combination of subject headings and keywords related to: respiratory tract infections including COVID-19 and other coronaviruses; swabs related to the oral and nasal cavities, nasopharynx and oropharynx; and aerosol generation and infection transmission. We did not limit the search to COVID-19 or coronavirus infection (see online supplemental appendix 1 for the search strategy).

We also applied the same search strategy as of 25 March 2020 to the China National Knowledge Infrastructure (CNKI) and Chinese Medical Journal Network (CMJN) (YC). We did not apply any language or quality restrictions. Finally, we searched medRxiv and ClinicalTrials. gov for articles related to COVID-19 or SARS-CoV-2 for preprints or grey literature addressing the research question.

### Selection criteria

We included all comparative and non-comparative studies that evaluated dispersion or aerosolisation of viable airborne organisms or transmission of infection associated with nasopharyngeal or oropharyngeal swab testing. We included studies that evaluated outcomes including detection of viable airborne organisms through microbiological sample analysis or documented transmission of infection associated with swab testing. We initially planned to include studies of hospitalised adult and paediatric patients with microbiologically confirmed COVID-19 in one or more fluid samples, of which at least one was a nasopharyngeal or oropharyngeal swab. In the absence of direct evidence meeting the eligibility criteria above, we broadened our inclusion criteria to include all study designs and populations evaluating aerosol generation or dispersion associated with swab testing. This included studies of hospitalised and non-hospitalised patients with or without microbiologically confirmed COVID-19, simulation studies without human participants, and those describing dispersion of non-infectious air particles or liquid droplets.

### Study selection

Paired reviewers (SM, KH, LB and SB) independently screened citations and conducted full-text review of all potentially eligible studies. Reviewers screened the reference lists of articles to identify additional studies meeting eligibility criteria, and screened preprints and grey literature from medRxiv and ClinicalTrials.gov. Disagreements were resolved by a third reviewer (AA) at all stages. Paired reviewers (YC, XY, NY and XL) screened citations identified from the CNKI and CMJN and resolved disagreements by discussion.

### Data extraction and quality assessment

Paired reviewers planned to abstract data using a standardised form. We planned to assess risk of bias (RoB) using the modified Cochrane RoB tool for randomised controlled trials,[9] the Risk Of Bias In Non-randomised Studies - of Interventions (ROBINS-I) tool for observational studies,[10] and content expert assessments for experimental or simulation-based studies. We planned to assess the overall certainty of the evidence based on RoB, imprecision, indirectness, inconsistency and publication bias, informed by Grading of Recommendations Assessment, Development and Evaluation (GRADE) guidance.[11 12]

### Data analysis

If insufficient data were available for pooled analyses, we planned to summarise results narratively.

## RESULTS

Of the 7702 citations identified in our search, 5477 were screened for eligibility after duplicates were removed. Of these, nine were considered potentially eligible requiring full-text review. After careful full-text review, we identified one observational study examining transmission rates associated with a safety protocol for nasopharyngeal swabs for detection of SARS-CoV-2.[13] This study implemented

and evaluated a safety protocol for nasopharyngeal swab sampling in a tertiary care hospital in Wuhan, China. The protocol involved establishment of a special sampling room with negative pressure ventilation, training of nursing staff conducting swab tests, strict sterilisation protocols, personal protective equipment including N95 or higher-level masks with standard droplet and contact precautions, and standardised swab collection and transportation methods. Based on preliminary evaluation among 8 nurses conducting over 11 000 nasopharyngeal swabs, a 0% nursing staff infection rate was reported.[13]

We did not apply the ROBINS-I tool, given the study was non-comparative. Overall certainty of evidence was very low, with very serious concerns regarding RoB, imprecision and publication bias.

No studies evaluating dispersion and aerosol generation associated with nasopharyngeal or oropharyngeal swabs, nor any studies comparing transmission using different safety protocols, were found despite broadening eligibility criteria (figure 1).

## DISCUSSION

This systematic review on aerosol generation and associated risk of infection transmission with nasopharyngeal and oropharyngeal swabs in the detection of SARS-CoV-2 yielded limited to no evidence. Only one small study evaluating a preliminary safety protocol for nasopharyngeal swabs was found, which showed that standardised protocols and airborne precautions were associated with no infections

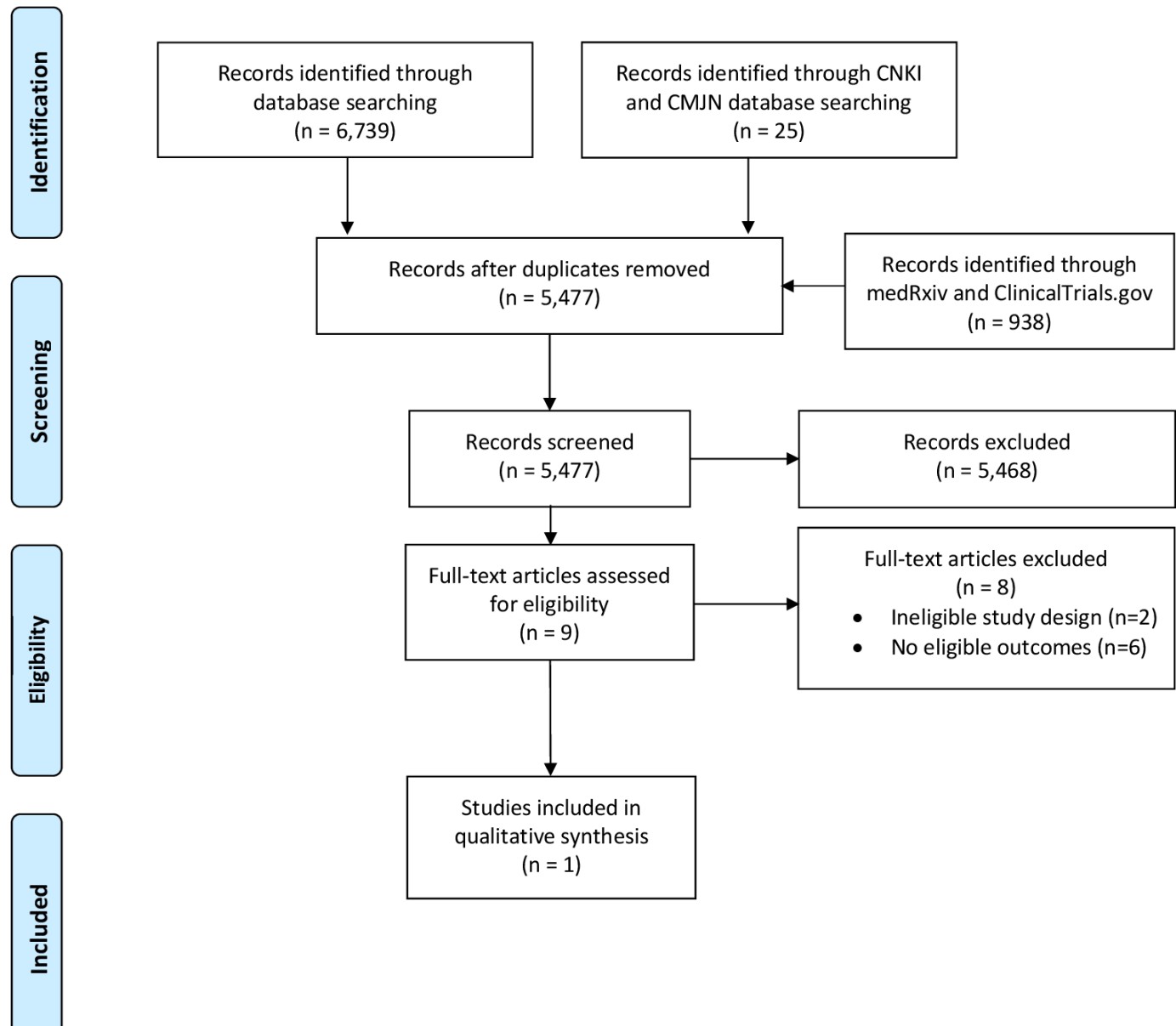

**Figure 1** PRISMA flow diagram for review on aerosol generation associated with nasopharyngeal and oropharyngeal swabs. CMJN, Chinese Medical Journal Network; CNKI, China National Knowledge Infrastructure; PRISMA, Preferred Reporting Items for Systematic Reviews and Meta-Analyses.

among nursing staff conducting testing.[13] The safety protocol implemented, which included a negative pressure room and N95 or higher masks, is not widely implemented in current practice; no data was provided regarding risk of transmission with nasopharyngeal swabs under other conditions. Broadening of eligibility criteria to allow for all study designs, including non-comparative and experimental designs and all populations regardless of COVID-19 infection, also did not yield any additional eligible studies.

Airborne transmission is different from droplet transmission, as it refers to the presence of microbes within droplet nuclei. These are generally considered to be particles <5 µm in diameter, which can remain in the air for periods of time, and can subsequently be transmitted to others over distances greater than 1 m.[6] Reduction of respiratory particles to <5 µm involves dehydration of larger droplets and consequent dehydration of any organisms within the droplet. If airborne particles are inhaled, they may rehydrate in the upper airway, increase in size and deposit in the airway. Thus, airborne transmission of viable organisms requires the organism to survive a process of desiccation and aerosolisation in sufficient quantity to incite infection. This property is specific to the organism.[14 15]

Aerosol generating procedures may expose healthcare workers and other contacts to pathogens, and the potential for transmission of respiratory infections. The risk of aerosolisation and transmission with swab testing remains unclear, particularly in the context of COVID-19. The most important theoretical risk associated with testing is induction of coughing with possible droplet release and aerosolisation. A previous study evaluating voluntary cough-generated aerosolisation of influenza virus using a bioaerosol cyclone sampler found that 81% of participants produced detectable viral RNA, with 23% of the RNA contained in particles 1–4 µm in size and 42% contained in particles less than 1 µm.[16]

Specific to COVID-19, experimental studies have shown aerosols with SARS-CoV-2 virus RNA in air samples for up to 3–16 hours[17 18]; however, studies from healthcare settings with symptomatic COVID-19 patients have yielded mixed results regarding identification of SARS-CoV-2 in air samples, and none have identified viable virus.[6]

Initial guidance from the WHO on 29 March 2020 recommended droplet and contact precautions for those people caring for COVID-19 patients, and airborne precautions for settings in which aerosol generating procedures and treatments are performed.[4] Subsequent guidance on 1 December 2020 recommended, in addition to standard droplet and contact precautions, that respirators, N95s or equivalent level masks be worn in care settings where such procedures are performed. Aerosol generating procedures were defined as 'tracheal intubation, non-invasive ventilation, tracheotomy, cardiopulmonary resuscitation, manual ventilation before intubation, bronchoscopy, sputum induction using nebulised hypertonic saline and dentistry and autopsy procedures'.[7] Guidance specific to nasopharyngeal and oropharyngeal swab testing were not explicitly provided.

Finally, recent evidence regarding salivary nucleic acid amplification testing has supported comparable diagnostic accuracy, with a pooled sensitivity of 83.2% (95% CIs 74.7% to 91.4%) and specificity of 99.2% (95% CI 98.2% to 99.8%), compared with nasopharyngeal swab testing (84.8% (95% CI 76.8% to 92.4%) and 98.9% (95% CI 97.4% to 99.8%), respectively).[19] While demonstrating comparable yield, salivary testing was associated with lower costs.[20] Coupled with increased ease of testing, reduced invasiveness and likely reduced occupational exposure risk, salivary testing may be a viable alternative to nasopharyngeal or oropharyngeal swab testing.

Taken together, recommendations regarding infection prevention and control measures during nasopharyngeal or oropharyngeal swab testing for SARS-CoV-2 should be carefully considered in light of the unknown risk of airborne transmission of infection to healthcare workers and other patients. Field experiments with nasopharyngeal and oropharyngeal swab testing in confirmed COVID-19 patients are required to adequately assess this risk using viral samplers, reverse transcriptase PCR testing and viral cultures. While we cannot estimate the risk of airborne transmission, certain factors may abrogate it: adequate room ventilation, limiting exposure to the patient, high filtration fit-tested respirators (eg, N95, FFP2) and low viral shedding.[13 21] However, the risks ought to be weighed against limited resource availability and the need for judicious use of personal protective equipment. One may also consider the yield of various fluid samples and balance their associated risks of aerosolisation with this consideration. The emergence of diagnostic tests such as salivary testing with less potential for aerosol generation may diminish the enthusiasm for nasopharyngeal and oropharyngeal swabs.

Strengths of this systematic review include a comprehensive literature search incorporating English and Chinese studies, as well as grey literature and preprints, and consultation with experts in the field. Limitations include a lack of published evidence informing the risk of aerosolisation with nasopharyngeal and oropharyngeal swab testing, including indirect evidence. This highlights an important gap in the existing literature, and precludes development of evidence-informed guidance beyond recommendations based on expert opinion.

## CONCLUSION

Our systematic review revealed limited to no data specifically addressing the risk of aerosol generation and airborne transmission associated with nasopharyngeal and oropharyngeal swab testing in the context of COVID-19 or related infectious respiratory illnesses. Current international guidelines recommend droplet and contact precautions during testing. Given a theoretical risk of cough-related aerosolisation during testing, field experiments to quantify the risk of aerosol generation and transmission associated with swab testing are warranted. In the interim, appropriate vigilance in adhering to available

standards for infection prevention and control measures is suggested. Alternative testing modalities, including salivary testing, may warrant consideration in place of nasopharyngeal and oropharyngeal testing.

**Author affiliations**
[1]Department of Medicine, University of Toronto, Toronto, Ontario, Canada
[2]Department of Health Research Methods, Evidence and Impact, McMaster University, Hamilton, Ontario, Canada
[3]Division of Critical Care, Department of Medicine, University of Ottawa, Ottawa, Ontario, Canada
[4]Department of Emergency Medicine, University of Ottawa, Ottawa, Ontario, Canada
[5]Division of Critical Care, Department of Medicine, Western University, London, Ontario, Canada
[6]Faculty of Science, McMaster University, Hamilton, Ontario, Canada
[7]School of Medicine and Biomedical Sciences, University of Buffalo, Buffalo, New York, USA
[8]Michael G. DeGroote School of Medicine, McMaster University, Hamilton, Ontario, Canada
[9]Department of Medicine, McMaster University, Hamilton, Ontario, Canada
[10]Department of Pediatrics, University of Toronto, Toronto, Ontario, Canada
[11]Division of Cardiology, Labatt Family Heart Centre, Hospital for Sick Children, Toronto, Ontario, Canada
[12]Center for Global Child Health, Hospital for Sick Children, Toronto, Ontario, Canada
[13]Department of Critical Care Medicine, Sunnybrook Health Sciences Centre, Toronto, Ontario, Canada
[14]Interdepartmental Division of Critical Care Medicine, University of Toronto, Toronto, Ontario, Canada
[15]Centre de recherche due CHU de Sherbrooke, Université de Sherbrooke, Sherbrooke, Quebec, Canada
[16]Département de Médecine, Université de Sherbrooke, Sherbrooke, Quebec, Canada
[17]Faculty of Medicine, BC Children's Hospital, University of British Columbia, Vancouver, British Columbia, Canada
[18]Department of Medicine and Therapeutics, Chinese University of Hong Kong, Shatin, Hong Kong
[19]Stanley Ho Center for Emerging Infectious Diseases, Chinese University of Hong Kong, Shatin, Hong Kong
[20]Department of Anaesthesia and Intensive Care, Chinese University of Hong Kong, Shatin, Hong Kong
[21]Division of Infectious Diseases, University of Toronto, Toronto, Ontario, Canada
[22]Department of Laboratory Medicine and Pathobiology, University of Toronto, Toronto, Ontario, Canada
[23]Pacific Medical Center, San Francisco, California, USA
[24]World Health Organization, Geneva, Switzerland
[25]Division of Critical Care, Unity Health Toronto – St. Michael's Hospital, Toronto, Ontario, Canada
[26]Li Ka Shing Knowledge Institute, Unity Health Toronto – St. Michael's Hospital, Toronto, Ontario, Canada
[27]Michael G. DeGroote Institute for Pain Research and Care, McMaster University, Hamilton, Ontario, Canada
[28]MAGIC Evidence Ecosystem Foundation, Oslo, Norway

**Acknowledgements** We would like to thank Yaolong Chen, Xuan Yu, Nan Yang and Xufei Luo for their assistance with systematic searching and screening of evidence from the China National Knowledge Infrastructure database for additional literature related to our systematic review. We would like to thank the World Health Organization for their support.

**Contributors** PV conceived the study. AA and PV organised the study teams and process. AA and RC designed the search strategy. SMF, KH, LB and SB screened studies for eligibility, and assessed study risk of bias and certainty of the body of evidence. AA addressed discrepancies in screening. AA wrote the first draft of the manuscript and conducted data analysis. AA, BR, NKJA, FL, SM, DSH, CDG, SM, JD, KEAB, JB, FM, DG, DiC, DeC, MH and PV critically revised the manuscript. AA and PV are the guarantors. All coauthors were involved in final editing of the manuscript, and approved the final draft. The corresponding author attests that all listed authors meet authorship criteria.

**Funding** This project was supported by the WHO (WHO Registration 2020/1010002-0) as part of a rapid review effort related to COVID-19, through the MAGIC Evidence Ecosystem Foundation (www.magicproject.org).

**Competing interests** None declared.

**Patient consent for publication** Not required.

**Provenance and peer review** Not commissioned; externally peer reviewed.

**Data availability statement** All data relevant to the study are included in the article or uploaded as online supplemental information. No additional data is available.

**ORCID iDs**
Arnav Agarwal http://orcid.org/0000-0002-0931-7851
Shannon M Fernando http://orcid.org/0000-0003-4549-4289
Kimia Honarmand http://orcid.org/0000-0002-7583-1445
David S Hui http://orcid.org/0000-0003-4382-2445

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
