## [Reviewer comments · BMJ Open]

ARTICLE DETAILS

TITLE (PROVISIONAL)	Risk of dispersion or aerosol generation and infection transmission with nasopharyngeal and oropharyngeal swabs for detection of COVID-19: a systematic review
AUTHORS	Agarwal, Arnav; Fernando, Shannon; Honarmand, Kimia; Bakaa, Layla; Brar, Sonia; Granton, David; Chaudhuri, Dipayan; Chetan, Devin; Hu, Malini; Basmaji, John; Muttalib, Fiona; Rochweg, Bram; Adhikari, Neill; Lamontagne, Francois; Murthy, Srinivas; Hui, David; Gomersall, Charles; Mubareka, Samira; Diaz, Janet; Burns, Karen; Couban, Rachel; Vandvik, Per

VERSION 1 – REVIEW

REVIEWER	Jonathan Luke Begley Cabrini Hospital, Australia
REVIEW RETURNED	23-Jun-2020

GENERAL COMMENTS	Thank you for giving me the chance to review your paper, which I enjoyed reading. It was disappointed that no papers could be found to answer the research question despite your extensive search (although I'm sure I was not as disappointed as you were!), and of course I agree that this provides justification for further research. I was impressed by the thorough search methodology including searching foreign-language and grey literature and agree that this is a strength of the study. The surrounding discussion was interesting and provides good context for the reader. My suggestions are minor: In discussion at page 12, lines 24-36: I really enjoyed this short description of how droplet nuclei are generated by the dehydration of larger droplets and the potential impact on organism viability. A quick search tells me this has been described for at least 40 years, so I am a little embarrassed that I didn't know of this mechanism! I suspect I won't be the only reader who isn't aware of this, and so I suggest that a reference would be beneficial. Cox 1989 'Airborne Bacteria and Viruses' in Science Progress (PMID 2699673) could be appropriate (I have no conflict of interest in suggesting this), or another suitable reference if you have a better one. Page 12, line 38 to Page 13, line 18 (paragraph beginning "Aerosol generating procedures may expose healthcare workers...") I think the structure of this paragraph may cause confusion. You discuss aerosol-generating procedures, and coughing being aerosol-generating. Then you talk about evidence that influenza is droplet-
---

	spread and the resulting WHO recommendations, before again talking about aerosols. My concern is that the transitions are subtle, and a casual reader may not notice the transition from discussing aerosols to droplets and back. Similarly, a casual reader may infer that experimental aerosol generation implies infectivity (even though you've stated elsewhere these are not the same thing). I suggest making clearer the distinction between aerosols and droplets in this paragraph, either by adding bridging phrases to define the transition and/or be rearranging the sentences. I also suggest making clearer that demonstrating RNA in aerosols does not necessarily imply infectivity, if you agree that it does not (Reference 5 discusses this in a very similar context, so could be cited again). Page 13, line 44: There is an error in this sentence ("The emergence of a diagnostic test with no potential for aerosol generation emerge (e.g. serology) may diminish the enthusiasm for nasopharyngeal and oropharyngeal swabs") — I suspect you intended to remove the word "emerge". References 4 and 13 have now had issue and page number allocated by the respective journals, so can be updated to their full citations. Figure 1: during conversion to PDF the arrows and/or boxes seem to have shifted so they're not properly aligned. Thanks for publishing this paper which hopefully will encourage much-needed research!
--	---

REVIEWER	Jingping Wang Massachusetts General Hospital, Boston, MA, USA
REVIEW RETURNED	28-Jun-2020

GENERAL COMMENTS	The authors provide a concise review of the risks of aerosols of nasopharyngeal and oropharyngeal swabs for COVID-19 from a very good perspective, given that the tests are now very common in COVID-19 screening in the world. The author has done a very detailed search and screening, although no citations about COVID-19 were eligible for inclusion. It is better to add some relevant evidence of throat swabs or nasal swabs for other respiratory viruses, such as seasonal influenza, SARS, MERS, etc., if there is any. Recommendations for infection prevention and control are given in this review, but are superficial and not as detailed as in recently published guidelines for other aerosol generation medical procedures. My suggestion is to specify and detail the protective measures and equipment, and forms or flowcharts are welcome.
---

REVIEWER	Shazia Jamil, MD Scripps Clinic, U.S.A University of California, San Diego School of Medicine, U.S.A
REVIEW RETURNED	20-Sep-2020

GENERAL COMMENTS	Reviewer's comments and recommendations: Excellent and timely systematic review of literature on the controversial topic of possibility of aerosolization of SARS CoV-2 during nasopharyngeal and oropharyngeal swabs techniques. I commend the authors for reviewing not only English language but
--

	Chinese language literature as well as pre-print literature as our understanding of this virus is evolving at an extremely fast rate. In addition, although authors initially started with only hospital testing they broadened to outpatient testing, which strengthens their review. As authors mentioned, although 6640 citations were identified, ONLY 6 were considered potentially eligible to be included in this review. Unfortunately, after careful review of full text, authors did not identify any citations evaluating aerosol generation related to nasopharyngeal or oropharyngeal swab testing. They also highlighted this point during the limitation of their study at the end of discussion. Therefore, I disagree and find it inaccurate that during their conclusion they state and I quote: There is no evidence regarding the risk of aerosol generation and airborne transmission associated with nasopharyngeal or oropharyngeal swab testing in the context of COVID-19 or related infectious respiratory illnesses. Absence of data or studies doesn't directly translate into absence of evidence but rather that we have been unable to accurately study it. I recommend that they revise the first statement of their Conclusion to state: Our extensive systematic review has not revealed any studies that have specifically addressed this question. There is a knowledge gap that exists in regards to understanding of aerosol transmission of SARS Co-V2 in humans during nasopharyngeal and oropharyngeal swab obtaining process. We need field human studies specifically to address this question. The rest of the statements under Conclusion are fine. Secondly, please replace COVID-19 with SARS Co-V2 terminology where applicable throughout the text for accuracy. The swabs are testing for the virus SARS Co-V2 which causes COVID-19 (the disease). For example, under Background, 2nd paragraph, 1st line, it states COVID-19 is primarily transmitted.....should be SARS Co-V2 is primarily transmitted. Likewise, under Background, last paragraph, last line should replace COVID-19 with SARS Co-V2. Under discussion, 1st paragraph, 2nd line, replace with detection of SARS Co-V2, 3rd paragraph replace as SARS Co-V2 has been shown to maintain stability..... I hope incorporation of above recommendations would strengthen the scientific rigor of this review.
--	--

VERSION 1 – AUTHOR RESPONSE

Reviewer: 1

Reviewer Name: Jonathan Luke Begley

Institution and Country: Cabrini Hospital, Australia

Please state any competing interests or state 'None declared': None declared

Comments to the Author

Dear Dr Agarwal,

Thank you for giving me the chance to review your paper, which I enjoyed reading.

It was disappointed that no papers could be found to answer the research question despite your extensive search (although I'm sure I was not as disappointed as you were!), and of course I agree that this provides justification for further research. I was impressed by the thorough search

methodology including searching foreign-language and grey literature and agree that this is a strength of the study. The surrounding discussion was interesting and provides good context for the reader.

Response: We thank the reviewer for sharing our enthusiasm on the topic, and our disappointment regarding the lack of high-quality evidence despite our comprehensive search. We appreciate the encouraging feedback.

My suggestions are minor:

In discussion at page 12, lines 24-36:

I really enjoyed this short description of how droplet nuclei are generated by the dehydration of larger droplets and the potential impact on organism viability. A quick search tells me this has been described for at least 40 years, so I am a little embarrassed that I didn't know of this mechanism! I suspect I won't be the only reader who isn't aware of this, and so I suggest that a reference would be beneficial. Cox 1989 'Airborne Bacteria and Viruses' in Science Progress (PMID 2699673) could be appropriate (I have no conflict of interest in suggesting this), or another suitable reference if you have a better one.

Response: We thank the reviewer for this comment. We agree references would be helpful to include here, and have included the suggested citation as well as a Pepper 2015 paper which summarizes this mechanism well (<https://doi.org/10.1016/B978-0-12-394626-3.00005-3>).

Page 12, line 38 to Page 13, line 18 (paragraph beginning "Aerosol generating procedures may expose healthcare workers...")

I think the structure of this paragraph may cause confusion. You discuss aerosol-generating procedures, and coughing being aerosol-generating. Then you talk about evidence that influenza is droplet-spread and the resulting WHO recommendations, before again talking about aerosols. My concern is that the transitions are subtle, and a casual reader may not notice the transition from discussing aerosols to droplets and back. Similarly, a casual reader may infer that experimental aerosol generation implies infectivity (even though you've stated elsewhere these are not the same thing).

I suggest making clearer the distinction between aerosols and droplets in this paragraph, either by adding bridging phrases to define the transition and/or be rearranging the sentences. I also suggest making clearer that demonstrating RNA in aerosols does not necessarily imply infectivity, if you agree that it does not (Reference 5 discusses this in a very similar context, so could be cited again).

Response: We thank the reviewer for this feedback. To provide more clarity here, we have omitted discussion regarding droplet-related WHO recommendations, and have kept the focus of the discussion to aerosol generation and the possibility of airborne transmission, which remains unclear, especially as it pertains to nasopharyngeal swabs.

Page 13, line 44: There is an error in this sentence ("The emergence of a diagnostic test with no potential for aerosol generation emerge (e.g. serology) may diminish the enthusiasm for nasopharyngeal and oropharyngeal swabs") — I suspect you intended to remove the word "emerge".

Response: Revised as suggested.

References 4 and 13 have now had issue and page number allocated by the respective journals, so can be updated to their full citations.

Response: We have updated the references.

Figure 1: during conversion to PDF the arrows and/or boxes seem to have shifted so they're not properly aligned.

Response: We thank the reviewer for this feedback. Formatting of Figure 1 has been fixed.

Thanks for publishing this paper which hopefully will encourage much-needed research!

Jonathan Begley

Response: We thank the reviewer for sharing our enthusiasm and for their support.

Reviewer: 2

Reviewer Name: Jingping Wang

Institution and Country: Massachusettes General Hospital, Boston, MA, USA

Please state any competing interests or state 'None declared': None declared

Comments to the Author

The authors provide a concise review of the risks of aerosols of nasopharyngeal and oropharyngeal swabs for COVID-19 from a very good perspective, given that the tests are now very common in COVID-19 screening in the world.

The author has done a very detailed search and screening, although no citations about COVID-19 were eligible for inclusion. It is better to add some relevant evidence of throat swabs or nasal swabs for other respiratory viruses, such as seasonal influenza, SARS, MERS, etc., if there is any.

Response: We thank the reviewer for this positive feedback. We have included both direct evidence pertaining to COVID-19, as well as indirect evidence pertaining to swabs in the context of other viral illnesses (such as influenza, SARS, MERS, etc); unfortunately, our search still yielded minimal evidence quantifying the risk of aerosol generation or dispersion, and related transmission.

Recommendations for infection prevention and control are given in this review, but are superficial and not as detailed as in recently published guidelines for other aerosol generation medical procedures. My suggestion is to specify and detail the protective measures and equipment, and forms or flowcharts are welcome.

Response: We thank the reviewer for this feedback. Given the lack of available evidence, we feel it is most appropriate to encourage appropriate vigilance and adherence to internationally available guidance, and this lack of data precludes more detailed recommendations.

Reviewer: 3

Reviewer Name: Shazia Jamil

Institution and Country: Scripps Clinic, U.S.A

Please state any competing interests or state 'None declared': NONE

Comments to the Author

Reviewer's comments and recommendations:

Excellent and timely systematic review of literature on the controversial topic of possibility of aerosolization of SARS CoV-2 during nasopharyngeal and oropharyngeal swabs techniques. I commend the authors for reviewing not only English language but Chinese language literature as well as pre-print literature as our understanding of this virus is evolving at an extremely fast rate. In addition, although authors initially started with only hospital testing they broadened to outpatient testing, which strengthens their review.

Response: We thank the reviewer for this positive feedback. We agree the literature is evolving at an extremely rapid rate. We have updated the English language search to keep the evidence base current in our revision.

As authors mentioned, although 6640 citations were identified, ONLY 6 were considered potentially eligible to be included in this review. Unfortunately, after careful review of full text, authors did not identify any citations evaluating aerosol generation related to nasopharyngeal or oropharyngeal swab testing. They also highlighted this point during the limitation of their study at the end of discussion. Therefore, I disagree and find it inaccurate that during their conclusion they state and I

quote: There is no evidence regarding the risk of aerosol generation and airborne transmission associated with nasopharyngeal or oropharyngeal swab testing in the context of COVID-19 or related infectious respiratory illnesses. Absence of data or studies doesn't directly translate into absence of evidence but rather that we have been unable to accurately study it.

Response: We thank the reviewer for this feedback. We have clarified our text to reflect that there is limited to no published evidence evaluating the topic in question.

I recommend that they revise the first statement of their Conclusion to state: Our extensive systematic review has not revealed any studies that have specifically addressed this question. There is a knowledge gap that exists in regards to understanding of aerosol transmission of SARS Co-V2 in humans during nasopharyngeal and oropharyngeal swab obtaining process. We need field human studies specifically to address this question. The rest of the statements under Conclusion are fine.

Response: We thank the reviewer for this feedback. We have amended the Conclusion to reflect this feedback; the revised sentence now reads: "Our systematic review revealed limited to no data specifically addressing the risk of aerosol generation and airborne transmission associated with nasopharyngeal and oropharyngeal swab testing in the context of COVID-19 or related infectious respiratory illnesses." We discuss the need for field studies addressing this question in our conclusion as well, in line with the reviewer's suggestion.

Secondly, please replace COVID-19 with SARS Co-V2 terminology where applicable throughout the text for accuracy. The swabs are testing for the virus SARS Co-V2 which causes COVID-19 (the disease). For example, under Background, 2nd paragraph, 1st line, it states COVID-19 is primarily transmitted.....should be SARS Co-V2 is primarily transmitted. Likewise, under Background, last paragraph, last line should replace COVID-19 with SARS Co-V2. Under discussion, 1st paragraph, 2nd line, replace with detection of SARS Co-V2, 3rd paragraph replace as SARS Co-V2 has been shown to maintain stability.....

Response: We thank the reviewer for this feedback regarding terminology. We have made the suggested changes throughout the manuscript for consistency and accuracy.

I hope incorporation of above recommendations would strengthen the scientific rigor of this review. Thank you

Response: We thank the reviewer for their valuable feedback and positive comments.

VERSION 2 – REVIEW

REVIEWER	Jonathan Begley Cabrini Hospital, Australia
REVIEW RETURNED	30-Jan-2021
GENERAL COMMENTS	Thank you again for giving me the chance to review this revision. In this revision the search has been updated which revealed one relevant study, which the authors correctly point out has some significant limitations. The fact that 12-months after the emergence of this virus there remains little evidence to guide the appropriate choice of protective equipment for performing diagnostic tests is a very important finding. Hopefully publication will encourage further research as the authors suggest. The paper is ready for publication.